# Inverse Preference Learning:
# Preference-based RL without a Reward Function

**Joey Hejna** [1]   **Dorsa Sadigh** [1]

## Abstract

Reward functions are difficult to design and often hard to align with human intent. Preference-based Reinforcement Learning (RL) algorithms address these problems by learning reward functions from human feedback. However, the majority of preference-based RL methods naïvely combine supervised reward models with off-the-shelf RL algorithms. Contemporary approaches have sought to improve performance and query complexity by using larger and more complex reward architectures such as transformers. Instead of using highly complex architectures, we develop a new and parameter-efficient algorithm, Inverse Preference Learning (IPL), specifically designed for learning from offline preference data. Our key insight is that for a fixed policy, the $Q$-function encodes all information about the reward function, effectively making them interchangeable. Using this insight, we completely eliminate the need for a learned reward function. Our resulting algorithm is simpler and more parameter-efficient. Across a suite of continuous control and robotics benchmarks, IPL attains competitive performance compared to more complex approaches that leverage transformer-based and non-Markovian reward functions while having fewer algorithmic hyperparameters and learned network parameters.

## 1. Introduction

Reinforcement Learning (RL) has shown marked success in fixed and narrow domains such as simulated control (Haarnoja et al., 2018) and game-playing (Mnih et al., 2013). When deploying RL in more complex settings, like in robotics or interaction with humans, one often runs into a critical bottleneck: the reward function. Obtaining reward labels in the real world can be complex, requiring

difficult instrumentation (Schenck & Fox, 2017; Zhu et al., 2020) and painstaking tuning (Yu et al., 2020) to achieve reasonable levels of sample efficiency. Moreover, despite extensive engineering, reward functions can still be exploited by algorithms in ways that do not align with human values and intents (Hadfield-Menell et al., 2017), which can be detrimental in safety-critical applications (Amodei et al., 2016).

Instead of hand-designing reward functions, contemporary works have attempted to learn them through expert demonstrations (Abbeel & Ng, 2004), natural language (Lin et al., 2022), or human feedback (Sadigh et al., 2017; Akrour et al., 2011; Wilson et al., 2012). Recently, reward functions learned through pairwise comparison queries—where a user is asked which of two demonstrated behaviors they prefer—have been shown to be effective in both control (Christiano et al., 2017; Sadigh et al., 2017; Lee et al., 2021a) and natural language domains (Stiennon et al., 2020). This is often referred to as *Reinforcement Learning with Human Feedback (RLHF)*. Reward functions learned via RLHF can directly capture human intent, while avoiding alternative and more expensive forms of human feedback such as expert demonstrations. Preference-based RL algorithms for RLHF often interleave reward-learning from comparisons with off-the-shelf RL algorithms.

While preference-based RL methods discover reward functions that are aligned with human preferences, they are not without flaws. Learned reward functions must have adequate coverage of both the state and action space to attain good downstream performance. Consequently, learning the reward function can be expensive, usually requiring thousands of labeled preference queries. To mitigate these challenges, recent works have proposed improving learned reward functions by adding inductive biases before optimization with RL. Hejna & Sadigh (2022) pretrain reward functions with meta-learning. Park et al. (2022) use data augmentation. Early et al. (2022) and Kim et al. (2023) make the reward function non-Markovian using recurrent or large transformer sequence model architectures respectively. Such approaches increase the upfront cost of preference-based RL by using additional data or compute. Moreover, these techniques still combine reward optimization with vanilla RL algorithms. Ultimately, this just adds an extra learned component to

[1]Stanford University. Correspondence to: Joey Hejna <jhejna@cs.stanford.edu>.

*Interactive Learning with Implicit Human Feedback Workshop at ICML 2023*

already notoriously delicate RL algorithms, further increasing hyper-parameter tuning overhead. Preference-based RL approaches often end up training up to four distinct neural networks independently: a critic (with up to two networks), an actor, and a reward function. This can be problematic as prediction errors cascade from the reward function, to the critic, and ultimately the actor causing high variance in downstream performance. To address these issues, we propose a parameter-efficient algorithm specifically designed for preference-based RL that completely eliminates the need to explicitly learn a reward function. In doing so, we reduce both complexity and compute cost.

The key insight of our work is that, under a fixed policy, the $Q$-function learned by off-policy RL algorithms captures the same information as the learned reward function. For example, both the $Q$-function and reward function encode information about how desirable a state-action pair is. This begs the question: why do we need to learn a reward function in the first place? Our proposed solution, Inverse Preference Learning or IPL, is an offline RL algorithm that is specifically designed for learning from preference data. Instead of relying on an explicit reward function, IPL directly optimizes the implicit rewards induced by the learned $Q$-function to be consistent with expert preferences. At the same time, IPL regularizes these implicit rewards to ensure high-quality behavior. As a result, IPL removes the need for a learned reward function and its associated computational and tuning expense.

Experimentally, we find that even though IPL does not explicitly learn a reward function, it achieves competitive performance with complicated Transformer-based reward learning techniques on offline Preference-based RL benchmarks with real-human feedback. At the same time, IPL consistently exhibits lower variance across runs as it does not suffer from the errors associated with querying a learned reward model. Finally, under a minimal parameter budget, IPL is able to outperform standard preference-based RL approaches that learn an explicit reward model.

## 2. Related Work

Our work builds upon literature in reward learning, preference-based RL, and imitation learning.

**Reward Learning.** Due to the challenges associated with designing and shaping effective reward signals, several works have investigated various approaches for learning reward functions. A large body of work uses inverse RL to learn a reward function from expert demonstrations (Abbeel & Ng, 2004; Ng et al., 2000; Ramachandran & Amir, 2007), which are unfortunately difficult to collect (Khurshid & Kuchenbecker, 2015; Akgun et al., 2012; Losey et al., 2020) or often misaligned with true human preferences (Basu et al.,

2017; Kwon et al., 2020). Subsequently, reward learning techniques using other simpler forms of feedback such as scalar scores (Knox & Stone, 2008) and partial (Myers et al., 2022) or complete rankings (Brown et al., 2019; Bıyık et al., 2019) have been developed. One of the simplest forms of human feedback is pairwise comparisons, where the user chooses between two options. Often, pairwise comparison queries are sampled using techniques from active learning (Sadigh et al., 2017; Biyik et al., 2020; Daniel et al., 2015). However, to evaluate learned reward functions, these methods rely on either RL or traditional planning algorithms which are complex and computationally expensive. Our approach takes a simpler perspective that is parameter-efficient by combining reward and policy learning. Though it is not the focus of our work, IPL could additionally leverage active learning techniques for selecting preference data online.

**Preference-based Deep Reinforcement Learning.** Current approaches to preference based deep RL train a reward function, and then use that reward function in conjunction with a standard reinforcement learning algorithm (Christiano et al., 2017; Lee et al., 2021b; Shin & Brown, 2021). Several techniques have been developed to improve the learned reward function, such as pre-training (Ibarz et al., 2018; Lee et al., 2021a), meta-learning (Hejna & Sadigh, 2022), data augmentation (Park et al., 2022), and non-Markovian modeling. Within the family of non-Markovian reward modeling (Bacchus et al., 1996), recent approaches have leveraged both LSTM networks (Early et al., 2022) and transformers (Kim et al., 2023) for reward learning. But, these methods still rely on Markovian offlien RL algorithms such as Implicit Q-Learning (IQL) (Kostrikov et al., 2022) for optimization. Ultimately, this makes such approaches theoretically inconsistent as the policy learning component assumes the reward to be only a function of the current state and action. All techniques for learning the reward function in combination with standard RL methods (Haarnoja et al., 2018; Schulman et al., 2017) end up adding additional hyper-parameter tuning and compute cost. IPL on the other hand, is directly designed for RL from preference data and eliminates the reward network entirely.

Recently, works in natural language processing have applied ideas from preference-based RL to tasks such as summarization (Stiennon et al., 2020; Wu et al., 2021), instruction following (Ouyang et al., 2022), and question-answering (Nakano et al., 2021). The RLHF paradigm has proven to be powerful even at the massive scale of aligning large language models. In this regime, learned reward models are massive, making an implicit reward method like IPL more attractive. While we focus on control in our experiments, we hope our work can inform future explorations in language domains.

**Imitation Learning**. Our work build on foundational

knowledge in maximum entropy (MaxEnt) RL (Ziebart, 2010) and inverse RL (Ziebart et al., 2008). Recent works in MaxEnt inverse RL have used the mapping between $Q$-functions and reward functions under a fixed policy. Specifically, Garg et al. (2021) show that the regularized MaxEnt inverse RL objective from Ho & Ermon (2016) can be rewritten using the $Q$-function instead of a reward function and Al-Hafez et al. (2023) stabilize their approach. While the relationship between $Q$-functions and rewards has been used for MaxEnt inverse RL, we study this relationship when learning from preference data. While both problems seek to learn models of expert reward, the data differs significantly — preference-based RL uses comparisons instead of optimal demonstrations. This necessitates a greatly different approach.

## 3. Inverse Preference Learning

In this section, we first describe the preference-based RL problem. Then, we describe how, leveraging techniques from imitation learning, we can remove the independently learned reward network from prior methods. This results in a simpler algorithm with lower computational cost and variance in performance.

### 3.1. Offline Preference-Based RL

We consider the offline reinforcement learning (RL) paradigm where an agent seeks to maximize its expected cumulative discounted sum of rewards in a Markov Decision Process (MDP) given an offline dataset $\mathcal{D}_o$ comprised of state, action, and next state tuples $(s, a, s')$ generated by an unknown behavior policy $\mu(a|s)$. However, unlike in the standard offline RL paradigm, in preference-based offline RL we do not assume access to reward labels in $\mathcal{D}_o$. Instead, the expert reward function $r_E(s, a)$ is unknown and must be learned from human feedback. Traditional preference-based RL methods are thus usually separated into two stages: first, reward learning, where $r_E$ is estimated, and second, reinforcement learning, where a policy $\pi(a|s)$ is learned to maximize $\mathbb{E}_\pi[\sum_{t=0}^\infty \gamma^t r_E(s, a)]$ with $\gamma$ as the discount factor. Though our method combines these two phases, we use the building blocks of each and consequently review them here.

First, similar to prior works (Christiano et al., 2017; Lee et al., 2021a), we assume access to preference data in the form of binary comparisons. Each comparison is comprised of two behavior segments, $\sigma^{(1)}$ and $\sigma^{(2)}$, and a binary label $y$ indicating which of the two was preferred by an expert. As in Wilson et al. (2012), each behavior segment is simply a snippet of a trajectory of length $k$, or $\sigma = (s_t, a_t, s_{t+1}, a_{t+1}, \ldots, a_{t+k-1}, s_{t+k})$. Increasing $k$ can provide more information per label at the cost of potentially noisier labels. The label $y$ for each comparison is assumed

to be generated by an expert according to a Bradley-Terry Preference model (Bradley & Terry, 1952):

$$P_E[\sigma^{(1)} > \sigma^{(2)}] = \frac{\exp \sum_t r_E(s_t^{(1)}, a_t^{(1)})}{\exp \sum_t r_E(s_t^{(1)}, a_t^{(1)}) + \exp \sum_t r_E(s_t^{(2)}, a_t^{(2)})}, \tag{1}$$

where $r_E(s_t, a_t)$ is again the expert's unknown underlying reward model. We use the subscript $E$ on probability $P$ to indicate that the preference distribution above results from the expert's reward function. Let the dataset of these preferences be $\mathcal{D}_p = \{(\sigma^{(1)}, \sigma^{(2)}, y)\}$. To learn $r_E$, prior works in preference-based RL estimate a parametric reward function $r_\theta$ by minimizing the binary-cross-entropy over $\mathcal{D}_p$:

$$\mathcal{L}_p(\theta) = \mathbb{E}_{\sigma^{(1)}, \sigma^{(2)}, y \sim \mathcal{D}_p} \left[ y \log P_\theta \left[ \sigma^{(1)} > \sigma^{(2)} \right] \right.$$
$$\left. + (1 - y) \log \left( 1 - P_\theta \left[ \sigma^{(1)} > \sigma^{(2)} \right] \right) \right]. \tag{2}$$

This objective results from simply minimizing $\mathbb{E}_{\mathcal{D}_p}[D_{\text{KL}}(P_E || P_\theta)]$, the KL-divergence between the expert preference model and the one induced by $r_\theta$, effectively aligning it with the expert's preferences. We note that some other works in preference-based RL focus on learning an improved model $r_\theta$ to address the reward learning part of the problem (Park et al., 2022; Kim et al., 2023). However, these methods still use off-the-shelf RL algorithms for the policy learning part of the problem.

Common approaches to offline RL seek to learn *conservative* policies that do not stray too far away from the distribution of data generated by $\mu(a|s)$. This is critical to prevent the policy $\pi$ from reaching out-of-distribution states during deployment which can be detrimental to performance. A common solution to this problem is to use a constrained objective (Kumar et al., 2020). In our derivations, we focus on the constrained objective from Garg et al. (2023):

$$\max_\pi \mathbb{E}_\pi \left[ \sum_{t=t'}^\infty \gamma^t \left( r(s_t, a_t) - \alpha \log \frac{\pi(a|s)}{\mu(a|s)} \right) \right]. \tag{3}$$

The second term subtracted from the reward $r$ enforces a KL-divergence constraint with the behavior policy $\mu$, encouraging the policy to remain near the dataset. Off-policy RL algorithms commonly use the contractive soft-Bellman operator,

$$(\mathcal{B}_r^\pi Q)(s, a) = r(s, a) + \gamma \mathbb{E}_{s' \sim p(\cdot|s|a)}[V^\pi(s')],$$
$$\text{where } V^\pi(s) = \mathbb{E}_{a \sim \pi(\cdot|s)} \left[ Q(s, a) - \alpha \log \frac{\pi(a|s)}{\mu(a|s)} \right], \tag{4}$$

for policy evaluation where $Q$ is the state-action value function and $V$ is the state value function. Recent works in offline RL have shown that the optimal $Q$-function, $Q^*$, for

Equation (3) can be attained directly using the optimal soft-Bellman operator $\mathcal{B}_r^*$ via the closed form optimal soft-value function $V^*(s) = \alpha \log \mathbb{E}_{a \sim \mu(\cdot|s)} \left[ e^{Q(s,a)/\alpha} \right]$ (Garg et al., 2023; Xu et al., 2023). After repeatedly applying $\mathcal{B}_r^*$, the policy can easily be extracted from $Q$.

To learn the optimal policy, two-phase preference based RL methods rely on recovering the optimal $r_E$ in the reward learning phase before running offline RL. This potentially propagates errors from the estimated $r_\theta$ to learned $Q$-function and ultimately learned policy $\pi$. In practice, it would be more efficient to eliminate the need for two separate stages. In the next section, we show how this can be done by establishing a bijection between reward functions $r$ and $Q$-functions.

### 3.2. Removing The Reward Function

In this section, we formally describe how the reward function can be removed from offline preference-based RL algorithms. Our key insight is that the $Q$-function learned by off-policy RL algorithms in fact encodes the same information as the reward function $r(s,a)$. Consequently, it is unnecessary to learn both. First, we show how the reward function can be re-written in terms of the $Q$ function allowing us to compute the preference model $P_Q$ induced by the $Q$-function. Then, we derive an objective that simultaneously pushes $Q$ to fit the expert's preferences while also remaining optimal.

Consider fitting a $Q$ function via the Bellman operator $\mathcal{B}_r^\pi$ for a fixed policy $\pi$ until convergence where $\mathcal{B}_r^\pi Q = Q$. Here, to encode the cumulative discounted rewards when acting according to the policy, the $Q$-function depends on both $r$ and $\pi$. This dependence, however, is directly disentangled by the Bellman equation. By rearranging it (Equation (4)), we can solve for the reward function in terms of $Q$ and $\pi$. This yields the so-called inverse soft-Bellman operator:

$$(\mathcal{T}^\pi Q)(s,a) = Q(s,a) - \gamma \mathbb{E}_{s'}[V^\pi(s')]. \quad (5)$$

In fact, it has been shown that for a fixed policy $\pi$, the soft inverse-Bellman operator is bijective, implying a one-to-one correspondence between the $Q$ function and the reward function (Garg et al., 2021). Intuitively, this makes sense: when holding the policy constant, only the reward function affects $Q$. We abbreviate the evaluation of $(\mathcal{T}^\pi Q)(s,a)$ as $r_{Q^\pi}(s,a)$ to indicate that $r_{Q^\pi}$ is the unique implicit reward function induced by $Q^\pi$. Prior works in imitation learning leverage the inverse soft-Bellman operator to measure how closely the implicit reward model $r_{Q^\pi}$ aligns with expert demonstrations. Our key insight is that this equivalence can also be used to directly measure how closely our $Q$ function aligns with the expert preference model *without ever directly learning $r$*.

Consider the Bradley-Terry preference model in Equation

(1). For a fixed policy $\pi$ and its corresponding $Q^\pi$, we can obtain the preference model of the implicit reward function $P_{Q^\pi}[\sigma^{(1)} > \sigma^{(2)}]$ by substituting the inverse soft-Bellman operator into Equation (1) as follows:

$$P_{Q^\pi}[\sigma^{(1)} > \sigma^{(2)}] = \frac{\exp \sum_t (\mathcal{T}^\pi Q)(s_t^{(1)}, a_t^{(1)})}{\exp \sum_t (\mathcal{T}^\pi Q)(s_t^{(1)}, a_t^{(1)}) + \exp \sum_t (\mathcal{T}^\pi Q)(s_t^{(2)}, a_t^{(2)})}. \quad (6)$$

This substitution will allow us to measure the difference between the preferences implied by $Q^\pi$ and those of the expert. To minimize the difference, we can propagate gradients through the preference modeling loss (Equation (2)) and the implicit preference model $P_{Q^\pi}$ (Equation (6)) to $Q$—just as we would for a parameterized reward estimate $r_\theta$. Unfortunately, naïvely performing the substitution is insufficient to solve the objective in Equation (3). This is because we have used an arbitrary policy $\pi$, not the optimal one, for converting from $Q$-values to rewards. Next, we show how we can form the optimal inverse soft-Bellman operator $\mathcal{T}^*$ to ensure the policy extracted from the learned $Q$-function is indeed optimal.

The optimal inverse soft-Bellman operator corresponds to the standard optimal soft-Bellman operator $\mathcal{B}_r^*$, which is formed by using the optimal soft-value function instead of $V^\pi(s)$ (Garg et al., 2023). Thus to construct $\mathcal{T}^*$ we substitute the form of $V$ used in $\mathcal{B}_r^*$ into Equation (5):

$$(\mathcal{T}^* Q)(s,a) = Q(s,a) - \gamma \mathbb{E}_{s'}[V^*(s')]$$
$$= Q(s,a) - \gamma \mathbb{E}_{s'}\left[\alpha \log \mathbb{E}_{a \sim \mu(\cdot|s)}\left[e^{Q(s,a)/\alpha}\right]\right].$$

Above we use the specific form of $V^*(s)$ for our KL-constrained objective, but in practice others can be used for different objectives. This makes our algorithm amenable to nearly any RL algorithm, like SAC (Haarnoja et al., 2018), so long as its value estimates converge to those of the desired policy. Thus in practice we simply update our estimates of $V$ according to any offline RL algorithm that converges to an optimal policy. For our KL-constrianed objective, this means fitting $V$ according to the linear exponential (linex) loss function from Extreme Q-Learning (XQL) (Garg et al., 2023). In Algorithm 3 we show this step written out. Thus, to fit the optimal $Q$-function, we minimize the following loss function:

$$\mathcal{L}_p(Q) = \mathbb{E}_{\sigma^{(1)}, \sigma^{(2)}, y \sim \mathcal{D}_p}\left[y \log P_{Q^*}[\sigma^{(1)} > \sigma^{(2)}]\right.$$
$$\left. + (1-y)\log(1 - P_{Q^*}[\sigma^{(1)} > \sigma^{(2)})\right].$$

Unfortunately, optimizing this objective alone leads to poor results due to the unconstrained nature of the implicit reward function. In practice, explicit reward learning approaches often constrain the learned reward function with either Tanh activations (Lee et al., 2021a) or normalization (Kim et al., 2023). These techniques, however, are inapplicable to our

*Figure 1.* A depiction of the difference between standard preference-based RL methods and Inverse Preference Learning. Standard preference-based RL first learns a reward function, then optimizes it with a blockbox RL algorithm. IPL trains a $Q$ function to directly fit the expert's preferences. This is done by aligning the implied reward model with the expert's preference distribution and applying regularization.

implicit algorithm. Instead, we introduce L2 regularization to the implicit reward, or $(r_{Q^*}(s, a))^2 = ((\mathcal{T}^*Q)(s, a))^2$, as is commonly done in imitation learning (Garg et al., 2021; Al-Hafez et al., 2023) to prevent unbounded reward values. We find that this also has a number of practical benefits. First, the solution to the Bradley-Terry preference model is non-unique, as any constant shift in all reward values does not change the probability of preferring any given segment. L2 regularization makes the solution unique and centers the induced reward near zero, which has shown to be beneficial for RL (Engstrom et al., 2020). Second, regularization encourages more realistic implicit rewards. For example, consider a reward function in continuous control that changes from -100 to 100 when one a small perturbation of size $\epsilon$ is applied to $s$ and $a$, but changes back to 100 when another perturbation of size $\epsilon$ is applied. While such a reward function seems unrealistic as it is unduly affected by small changes in states and actions, it is a completely valid solution of the inverse soft-Bellman operator. Adding L2 regularization discourages this discontinuous behavior by penalizing large deviations in implied reward unless they drastically reduce the preference loss. Finally, the benefits of implicit reward regularization naturally propagate to the learned $Q$-function. L2 regularization explicitly penalizes neighboring $Q$ and $V$ values from getting too far, smoothing the value landscape. This helps prevent exploding $Q$-values as estimates of $V$ typically increase throughout the course of learning. Thus, to encourage regularization across the entire state and action space, we use the following regularization objective over $Q$ which uses data from both $\mathcal{D}_p$ and $\mathcal{D}_o$:

$$\mathcal{L}_r(Q) = \mathbb{E}_{s,a \sim \mathcal{D}_o \cup \mathcal{D}_p} \left[ ((\mathcal{T}^*Q)(s, a))^2 \right].$$

Our final $Q$-learning objective thus the sum of the preference loss function under $\mathcal{T}^*Q$ and the regularization loss, or

$\mathcal{L}(Q) = \mathcal{L}_p(Q) + \lambda \mathcal{L}_r(Q)$, where $\lambda$ is a hyper-parameter that controls the regularization strength. $\mathcal{L}_p(Q)$ encourages the $Q$ function to implicitly match the the expert's preference model under the optimal policy. $\mathcal{L}_r(Q)$ makes the solution unique and smooths it. We find that weighting the regularization equally between $\mathcal{D}_p$ and $\mathcal{D}_o$ performs well. Note that $\mathcal{L}_p(Q)$ is only computed on preference data $\mathcal{D}_p$. In practice, optimizing $\mathcal{L}_p(Q)$ requires estimating $(\mathcal{T}^*Q)$ and consequently $V^*$. Thus, during optimization, we alternate between updating our learned $Q$-network and a learned value $V$-network towards $V^*$. After $Q$ and $V$ have converged, we can extract the policy using the closed form relationship $\pi^*(a|s) \propto \mu(a|s) \exp\left((Q^*(s, a) - V^*(s))/\alpha\right)$ for KL-constrained RL as in Garg et al. (2023); Peng et al. (2019).

---

**Algorithm 1** IPL Algorithm (XQL Variant)

**Input :** $\mathcal{D}_p, \mathcal{D}_o, \lambda, \alpha$
**for** $i = 1, 2, 3, ...$ **do**
    Sample batches $B_p \sim \mathcal{D}_p, B_o \sim \mathcal{D}_o$
    Update $Q$: $\min_Q \mathbb{E}_{B_p}[\mathcal{L}_p(Q)] + \lambda \mathbb{E}_{B_p \cup B_o}[\mathcal{L}_r(Q)]$
    Update $V$: $\min_V \mathbb{E}_{B_p \cup B_o}[e^z - z - 1]$
    where $z = Q(s, a) - V(s))/\alpha$

Finally, extract $\pi(a|s)$ via:
$\max_\pi \mathbb{E}_{\mathcal{D}_p \cup \mathcal{D}_o}[e^{(Q(s,a)-V(s))/\alpha} \log \pi(a|s)]$

---

We find that in the presence of additional offline data in $\mathcal{D}_o$, updating the policy and/or value function with data from both $\mathcal{D}_o$ and $\mathcal{D}_p$ performs better, just like with the regularization term. Ultimately, our optimization scheme learns a $Q$ function for an optimal policy whose implied reward is consistent with $r_E$, up to regularization, without ever learning the reward network. Practically, this has a number of

benefits. Learning a reward network requires more parameters and a completely separate optimization loop, increasing compute requirements. Moreover, an explicit reward model introduces a whole new suite of hyper-parameters that need to be tuned including the model architecture, capacity, learning rate, batch size, and stopping criterion. In fact, because human preference data is so difficult to collect, many approaches opt to use simple accuracy thresholds instead of validation criteria to decide when to stop training $r_\theta$ (Lee et al., 2021a). All of these components make preference-based RL unreliable and high-variance. On the other hand, our method completely removes all of these parameters in exchange for a single $\lambda$ hyper-parameter that controls the regularization strength. Despite removing these components, in the next section we will experimentally demonstrate that IPL performs just as well as explicit reward approaches.

# 4. Experiments

In this section, we aim to answer the following questions: First, how does IPL compare to prior preference-based RL algorithms on standard benchmarks? Second, how does IPL perform in extremely data-limited settings? And finally, how efficient is IPL in comparison to two-phase preference-based RL methods?

## 4.1. Setup

As discussed in the previous section, though we use a KL-constrained objective for our theoretical derivation, in practice we can construct versions of IPL based on any offline RL algorithm. In our experiments we evaluate IPL with Implicit Q-Learning (IQL) (Kostrikov et al., 2022), since it has been used in prior offline preference-based RL works. This allows us to directly compare IPL by isolating its implicit reward component and using the same exact hyper-parameters as prior works. Using IPL with IQL amounts to updating the value function according to the asymmetric expectile loss function instead of the linex loss function. Concretely, this can be done by replacing the value update in Algorithm 3 with $\min_V \mathbb{E}_{B_p \cup B_o} \left[ |\tau - \mathbb{1}(Q(s,a) - V(s))| (Q(s,a) - V(s))^2 \right]$ where $\tau$ is the expectile.

Inspired by Park et al. (2022), we introduce data augmentations that sample sub-sections of behavior segments $\sigma$ during training. While such augmentations are inapplicable to non-Markovian reward models, we find that they boost performance for Markovian reward models while also reducing the total number of state-action pairs per batch of preference data. This is important as IPL needs data from both $\mathcal{D}_p$ and $\mathcal{D}_o$ to regularize the implicit reward function. Additional experiment details and hyper-parameters can be found in the Appendix.

## 4.2. How does IPL perform on preference-based RL benchmarks?

We compare IPL to other offline preference-based RL approaches on D4RL Gym Locomotion (Fu et al., 2020) and Robosuite robotics (Mandlekar et al., 2021) datasets with real-human preference data from Kim et al. (2023). We compare IQL-based IPL, with the same hyper-parameters, to various baselines that learn a reward model $r_\theta$ before optimization with IQL. Markovian Reward or MR denotes using a standard Markovian MLP reward model, like those used in Christiano et al. (2017) and Lee et al. (2021a). Non-Markovian Reward or NMR denotes using the non-Markovian LSTM based reward model from Early et al. (2022). Preference Transformer (PT) is a state-of-the-art approach that leverages a large transformer architecture to learn a non-Markovian reward and preference weighting function. Finally, we compare against our own implementation of IQL with a Markovian Reward function that use the same data augmentation as IPL.

Our results are summarized in Table 1. Starting with the first column, we see that preference-based RL methods are able to match IQL with the ground truth reward function in many cases. On, several tasks, however, the MR implementation from Kim et al. (2023) fairs rather poorly. The non-Markovian methods, (NMR and PT) improve performance. It is worth noting that on many tasks our implementation of a MR (fifth column) performs far better than reported in Kim et al. (2023), likely due to our careful tuning of $r_\theta$ and use of data-augmentations. Our method, IPL, achieves competitive performance across the board.

In general, IPL performs on-par or better than both our implementation of MR and PT in most datasets despite not learning a separate reward network. Specifically, IPL has the same performance or better performance than our MR implementation on six of eight tasks. More importantly, IPL does extremely well in comparison to Preference Transformer's reported results. On five of eight tasks IPL performs better than PT while having over 10 times fewer parameters, making IPL far more efficient. To be consistent with Kim et al. (2023), we report results after a million training steps but performance for IPL often peaks earlier (see learning curves in the Appendix). For example, with early stopping IPL also outperforms PT on "hop-m-r". We posit that this is because the $Q$-function in IPL is tasked with both fitting the expert's preference model and optimal policy simultaneously, making both the policy and reward function non-stationary during training. In some datasets, this was more unstable.

IPL also has the lowest average standard-deviation across seeds, meaning it yields more consistent results than explicit reward methods. For standard two-phase preference-based RL algorithms, errors in the reward model are propagated

| Dataset | IQL (Oracle) | MR (from Kim et al.) | LSTM (from Kim et al.) | PT (from Kim et al.) | MR (reimpl.) | IPL (Ours) |
|---|---|---|---|---|---|---|
| hop-m-r | 83.06 ± 15.80 | 11.56 ± 30.27 | 57.88 ± 40.63 | **84.54** ± 4.07 | 70.20 ± 35.0 | 73.57 ± 6.67 |
| hop-m-e | 73.55 ± 41.47 | 57.75 ± 23.70 | 38.63 ± 35.58 | 68.96 ± 33.86 | **102.97** ± 5.55 | 74.52 ± 10.11 |
| walk-m-r | 73.11 ± 8.07 | 72.07 ± 1.96 | **77.00** ± 3.03 | 71.27 ± 10.30 | 68.79 ± 5.64 | 59.92 ± 5.11 |
| walk-m-e | 107.75 ± 2.02 | 108.32 ± 3.87 | **110.39** ± 0.93 | 110.13 ± 0.21 | 109.07 ± 1.30 | 108.51 ± 0.60 |
| lift-ph | 96.75 ± 1.83 | 84.75 ± 6.23 | 91.50 ± 5.42 | 91.75 ± 5.90 | **98.84** ± 2.33 | 97.60 ± 2.94 |
| lift-mh | 86.75 ± 2.82 | **91.00** ± 2.82 | **90.75** ± 5.75 | 86.75 ± 5.95 | 90.04 ± 4.45 | 87.20 ± 5.31 |
| can-ph | 74.50 ± 6.82 | 68.00 ± 9.13 | 62.00 ± 10.90 | 69.67 ± 5.89 | **76.40** ± 3.67 | 74.8 ± 2.40 |
| can-mh | 56.25 ± 8.78 | 47.50 ± 3.51 | 30.50 ± 8.73 | 50.50 ± 6.48 | 53.6 ± 7.86 | **57.6** ± 5.00 |
| Avg Std | 10.95 | 10.2 | 13.87 | 9.08 | 8.23 | **4.77** |

*Table 1.* Average normalized scores of all baselines on human-preference benchmarks from Kim et al. (2023). For the D4RL locomotion tasks "hop" corresponds to hopper, "m" to medium (training the data generating agent to 1/3 expert performance), "r" to replay buffer data, and "e" to data from the end of training. For the Robomimic tasks lift and can, "ph" corresponds to proficient human data and "mh" to multi-human data of differing optimality. The first four columns are taken from Kim et al. (2023). "reimpl." is our reimplementation of Markovian Reward with IQL. The "Avg Std" row shows the average standard deviation across all eight environments. We run five seeds and report the final performance at the end of training like Kostrikov et al. (2022). On some tasks IPL achieves higher performance earlier in training, which is not reflected above (See Appendix). We find that IPL outperforms PT on many environments, and also performs similarly to our implementation of MR despite not training a reward function.

to and exacerbated by the $Q$ function. IPL circumvents this problem by not explicitly learning the reward.

### 4.3. How does IPL scale with Data?

Collecting preference comparisons is often viewed as the most expensive part of preference-based RL. To investigate how well IPL performs in data limited settings, we construct scripted preference datasets of four different sizes for five tasks from the MetaWorld benchmark (Yu et al., 2020) used in prior Preference-based RL works (Lee et al., 2021a; Hejna & Sadigh, 2022). We then train on the preference data $\mathcal{D}_p$ by setting $\mathcal{D}_o = \{(s, a, s') \in \mathcal{D}_p\}$ and use the same hyper-parameters for all environments and methods where applicable. Our results are summarized in Table 2. Again, IPL is a strong reward-free baseline. We find that at all data scales, IPL performs competitively to our implementation of MR (IQL with a learned Markovian reward) and consistently outperforms it in Button Press and Assembly. Increasing the amount of preference data generally improves performance across the board. However, as we generate queries uniformly at random some preference datasets may be easier to learn from than others, leading to deviations form this trend in some cases. As in the benchmark results in Table 1, IPL exhibits lower variance across seeds and tasks, in this case at three of four data scales.

### 4.4. How efficient is IPL?

One benefit of IPL over other Preference-based RL methods is its parameter efficiency. By removing the reward network, IPL uses fewer parameters than other methods while achiev-

| Preference Queries | | 500 | 1000 | 2000 | 4000 |
|---|---|---|---|---|---|
| Button Press | MR | **66.0** ± 8.0 | 49.3 ± 12.1 | 54.7 ± 26.8 | 78.3 ± 9.2 |
| | IPL | 53.3 ± 8.5 | 60.1 ± 12.8 | 70.2 ± 2.5 | 90.2 ± 6.5 |
| Drawer Open | MR | 65.9 ± 9.9 | **87.2** ± 5.2 | 89.7 ± 6.4 | 94.6 ± 3.9 |
| | IPL | 62.1 ± 4.8 | 78.7 ± 12.4 | 89.5 ± 5.0 | 96.6 ± 1.3 |
| Sweep Into | MR | 33.0 ± 5.7 | 46.2 ± 6.0 | **63.2** ± 13.7 | **70.8** ± 7.9 |
| | IPL | 34.5 ± 2.3 | 48.2 ± 7.2 | 58.8 ± 7.4 | 65.9 ± 6.7 |
| Plate Slide | MR | **54.6** ± 5.3 | 57.2 ± 4.5 | 23.9 ± 18.8 | 55.2 ± 3.0 |
| | IPL | 52.9 ± 4.8 | 55.8 ± 2.2 | 55.4 ± 3.1 | 54.9 ± 2.8 |
| Assembly | MR | 0.6 ± 0.7 | 0.7 ± 1.0 | 0.0 ± 0.0 | 2.6 ± 2.8 |
| | IPL | 0.9 ± 0.6 | 1.5 ± 1.5 | 1.7 ± 1.9 | 5.5 ± 5.2 |
| Avg Std | MR | 5.9 | **5.76** | 13.14 | 5.36 |
| | IPL | **4.2** | 7.22 | **3.98** | **4.5** |

*Table 2.* Results on five MetaWorld tasks at four different preference data scales. We run five seeds for each method, and more details can be found in the Appendix. IPL performs the same or better than IQL with a Markovian reward model on the majority of tasks and preference data scales without training a reward model.

ing the same performance. In Table 3, we show the number of parameters for each method used in the last two sections. Preference Transformer uses over ten times more parameters than IPL, and the LSTM-based NMR model from Early et al. (2022) uses nearly twice as many. When dealing with a limited compute or memory budget, this can be important. To exacerbate this effect, we consider an extremely parameter efficient version of IPL, denoted "IPL (64)" in Table 3, based on Advantage Weighted Actor Critic (AWAC) (Nair et al., 2021) which eliminates the second critic and value networks used in IQL (Kostrikov et al., 2022) and uses a two-layer 64-dimensional MLP. We then compare this parameter-efficient IPL to MR with the same parameter

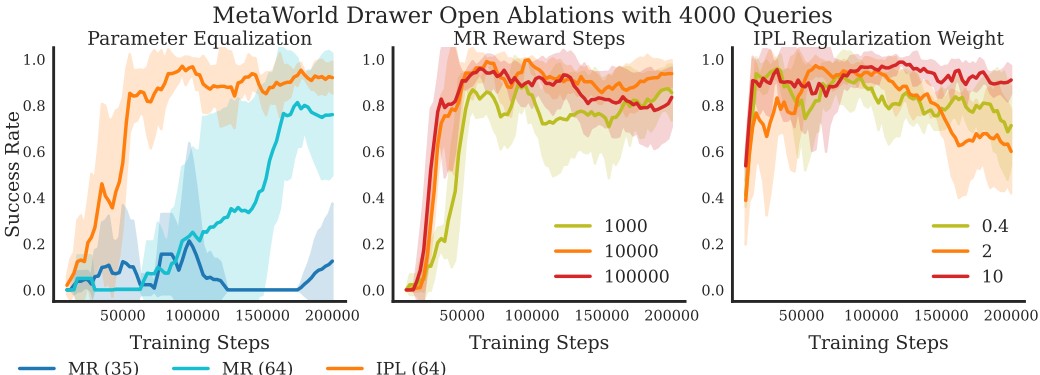

*Figure 2.* **Left:** Performance comparison with different parameter numbers. MR (35) has the same parameter budget as IPL (64). MR (64) has over twice as many. We see that with the same number of parameters as IPL, MR is unable to adequtely fit the data and performs poorly. **Middle:** MR when the reward function is trained for a varying number of steps – with too few the reward model under-fits, and with too many it over-fits, both leading to worse performance. **Right:** IPL with different regularization strengths. On the drawer open task, performance is largely unaffected. For more ablations, see the Appendix.

| Method | Params |
|---|---|
| PT | 2942218 |
| NMR | 508746 |
| MR | 348426 |
| IPL | 278537 |
| MR (64) | 34892 |
| IPL (64) | 14025 |
| MR (35) | 14012 |

*Table 3.* Parameter counts for different methods. The bottom three rows are for the limited parameter budget experiments in Section 4.4.

budget which results in "MR (35)", a 35-dimensional MLP. Results are depicted on the left of Figure 2. MR trained with a smaller network is unable to adequately fit the data, resulting in lower performance. Only after increasing the network size past that of IPL can MR begin to match performance.

Aside from parameter efficiency, IPL is also "hyper-parameter efficient". By removing the reward network, IPL removes a whole set of hyper-parameters associated with two phase preference based RL methods, like reward network architecture, learning rate, stopping criterion, and more. In the middle of Figure 2 we show how the performance of MR is affected when the reward function is over or under fit. Choosing the correct number of steps to train the reward model usually requires collecting a validation set of preference data, which is costly to obtain. Instead of this, IPL only has a single regularization parameter, $\lambda$. The right side of Figure 2 shows the sensitivity of IPL to $\lambda$. We find that in many cases, varying $\lambda$ has little effect on performance unless it is perturbed by a large amount. Due to space constraints, extended results for this section are included in the Appendix.

## 5. Conclusion

**Summary.** We introduce Inverse Preference Learning, a novel algorithm for offline preference-based RL that avoids learning a reward function. Our key insight is to leverage the inverse soft-Bellman operator, which computes the mapping from $Q$-functions to rewards under a fixed policy. The IPL algorithm trains a $Q$-function to regress towards the optimal $Q^*$ while at the same time admitting implicit reward values that are consistent with an expert's preferences. Even though IPL does not require learning a separate reward network, on robotics benchmarks it attains competitive performance with preference-based RL baselines that use twice to ten-times the number of model parameters.

**Limitations and Future Work.** A number of future directions remain. Specifically, the implicit reward function and policy learned by IPL are both non-stationary during training, which sometimes causes learning to be more unstable than with a fixed reward function. This is a core limitation future work could address by better mixing policy improvement and preference-matching steps to improve stability. More broadly, implicit reward preference-based RL methods are not limited to continuous control or binary feedback. Applying implicit reward techniques to other forms of feedback or extending IPL to language-based RLHF tasks remain exciting future directions.

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

# Appendix

We divide the appendix into four different sections following the results section. Each section additionally provides hyper-parameters used for IPL in that section. The first section, setup, contains details information on the experimental setup and hyper-parameters used. The second section on benchmark results gives full learning curves for the experiments in Section 4.2. The third section provides full learning curves for the MetaWorld and Data-scaling experiments. The final Appendix section provides extended ablations.

## A. Setup

Here we provide the full algorithmic outline of IPL using Implicit Q-Learning (Kostrikov et al., 2022) that mimics our implementation. While in practice the policy $\pi$ could be extracted at the end of training, we do it simultaneously as in (Kostrikov et al., 2022) in order to construct learning curves.

---

**Algorithm 2** IPL Algorithm (IQL Variant)

---

**Input :** $\mathcal{D}_p, \mathcal{D}_o, \lambda, \alpha$
**for** $i = 1, 2, 3, ...$ **do**
    Sample batches $B_p \sim \mathcal{D}_p, B_o \sim \mathcal{D}_o$
    Update $Q$: $\min_Q \mathbb{E}_{B_p}[\mathcal{L}_p(Q)] + \lambda \mathbb{E}_{B_p \cup B_o}[\mathcal{L}_r(Q)]$
    Update $V$: $\min_V \mathbb{E}_{B_p \cup B_o}\left[|\tau - \mathbb{1}(Q(s,a) - V(s))|(Q(s,a) - V(s))^2\right]$
    Update $\pi$: $\max_\pi \mathbb{E}_{\mathcal{D}_p \cup \mathcal{D}_o}[e^{\beta(Q(s,a) - V(s))} \log \pi(a|s)]$

---

Note that above we write the temperature parameter $\beta$ as done in IQL, instead of how it is usually done, using $\alpha$ in the denominator (*Garg et al.*, 2023; *Peng et al.*, 2019).

When sampling batches of preference data $B_p \sim \mathcal{D}_p$, we take sub-samples of each segment $\sigma$ of length $s$. For a sampled data point $(\sigma^{(1)}, \sigma^{(2)}, y)$, we sample start $\sim \text{Unif}[0, 1, 2, ...k - s]$ and then let take $\sigma = s_{\text{start}}, a_{\text{start}}, ..., s_{\text{start}+s}$. We use the same start value across the entire batch.

Given that we run experiments using MLPs, all of our experiments were run on CPU compute resources. Each seed for each method requires one CPU core (two hyper-threads) and 8 Gb of memory.

## B. Benchmark Results

Here we provide details for our experiments on the preference-based RL benchmark from Kim et al. (2023). We use the same hyperparameters as Kim et al. (2023) and Kostrikov et al. (2022) where applicable as shown in Table 4.

**Gym-Mujoco Locomotion**. Hopper and Walker2D agents are tasked with learning locomotion policies from datasets of varying qualities taken from the D4RL (Fu et al., 2020) benchmark. Preference datasets were constructed by Kim et al. (2023) by uniformly sampling queries and labeling a subset of them. For all locomotion tasks the segment length. Preference datasets for "medium" quality offline datasets contain 500 queries, while preference datasets for "expert" quality offline datasets contain 100 queries. Segment lengths $k = 100$ for all datasets, and were subsampled to length $s = 64$ by IPL and our MR (reimplementation). Evaluation was preformed over 10 episodes every 5000 steps. Full learning curves are shown in Figure 3.

**RoboMimic**. The RoboMimic datasets contain interaction data of two types: ph — proficient human and mh – multihuman. The multi-human data was collected from human demonstrators of mixed quality. The robot is tasked with learning how to lift a cube (lift) or pick and place a can (can). Preference datasets were again taken directly from Kim et al. (2023). Preference datasets of size 100 with segment lengths $k = 50$, randomly sub-sampled to length $s = 32$ were used for the ph datasets. Preference datasets of size 500 with segment lengths $k = 100$, randomly sub-sampled to length $s = 64$ were used for the mh datasets. Evaluation was performed over 25 episodes every 50000 steps. Full learning curves are shown in Figure 4.

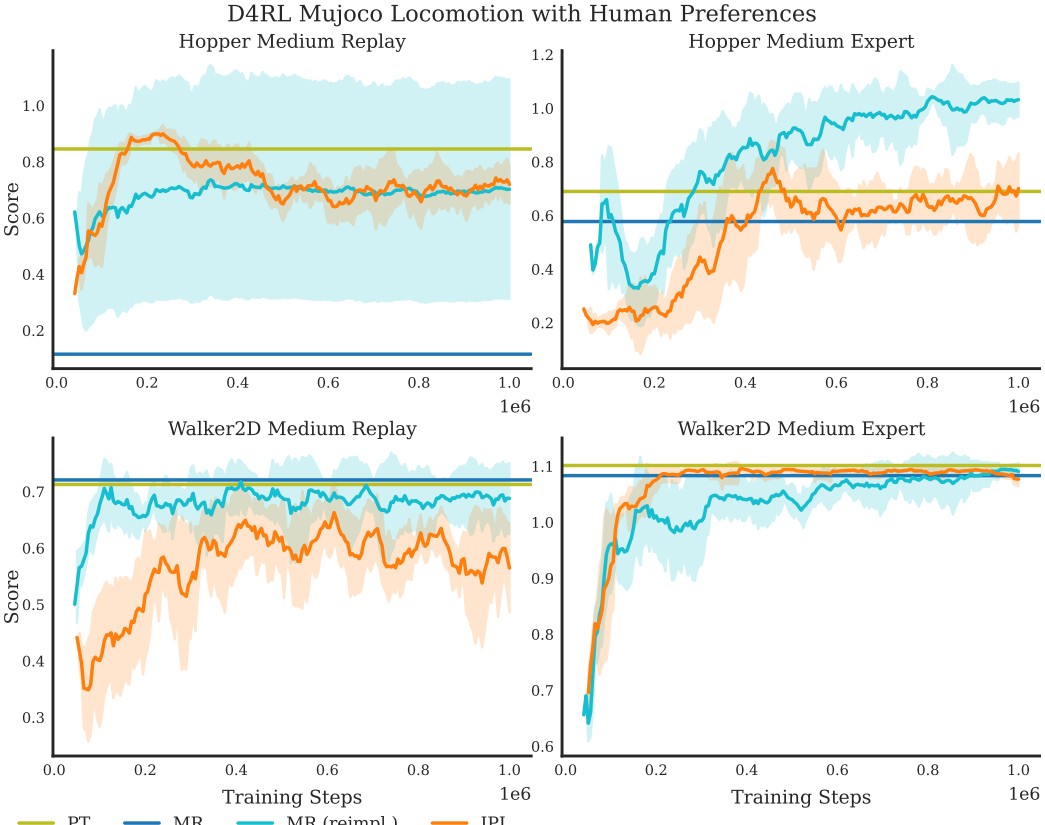

*Figure 3.* Full learning curves on the D4RL locomotion benchmark with human preferences.

| Common Hyperparameters | | |
|---|---|---|
| **Parameter** | **Locomotion** | **Robomimic** |
| $Q, V, \pi$ Arch | 2x 256d | 2x 256d |
| Learning Rate | 0.0003 | 0.0003 |
| Optimizer | Adam | Adam |
| $\beta$ | 3.0 | 0.5 |
| $\tau$ | 0.7 | 0.7 |
| $\mathcal{D}_o$ Batch Size | 256 | 256 |
| $\mathcal{D}_p$ Batch Size | 8 | 8 |
| Training Steps | 1 Mil | 1 Mil |
| $k$ | 100 | 100, 50 |
| Subsample $s$ | 64 | 64, 32 |

| MR Hyperparameters | | |
|---|---|---|
| **Parameter** | **Locomotion** | **Robomimic** |
| $r_\theta$ Arch | 2x 256d | 2x 256d |
| $r_\theta$ LR | 0.0003 | 0.0003 |
| $r_\theta$ Optimizer | Adam | Adam |
| $r_\theta$ Steps | 20k | 20k |

| IPL Hyperparameters | | |
|---|---|---|
| **Parameter** | **Locomotion** | **Robomimic** |
| $\lambda$ | 0.5 | 4 |

*Table 4.* Hyperparameters used for the benchmark experiments. We can see that IPL has fewer hyperparameters.

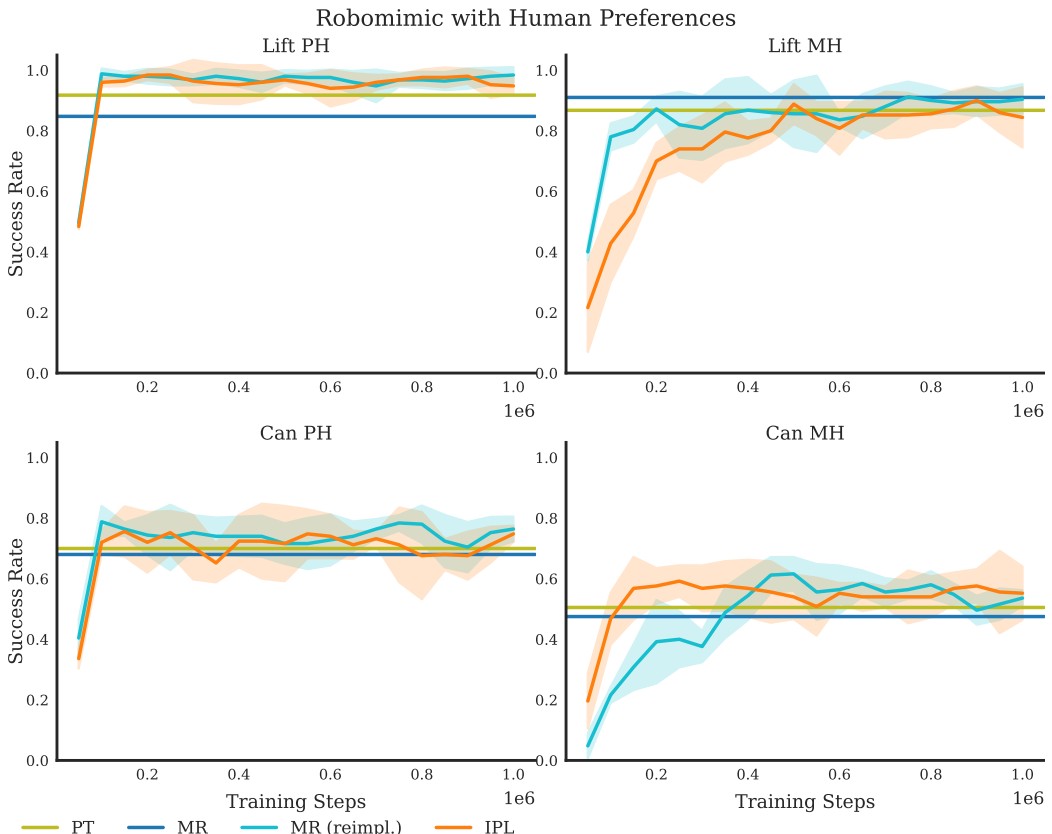

*Figure 4.* Full learning curves on the RoboMimic benchmark with human preferences.

| Common Hyperparameters | | | MR Hyperparameters | |
|---|---|---|---|---|
| **Parameter** | **Value** | | **Parameter** | **Value** |
| $Q, V, \pi$ Arch | 3x 256d | | $r_\theta$ Arch | 3x 256d |
| Learning Rate | 0.0003 | | $r_\theta$ LR | 0.0003 |
| Optimizer | Adam | | $r_\theta$ Optimizer | Adam |
| $\beta$ | 4.0 | | $r_\theta$ Steps | 20k |
| $\tau$ | 0.7 | | | |
| $\mathcal{D}_p$ Batch Size | 16 | | | |
| Training Steps | 200k | | **IPL Hyperparameters** | |
| $k$ | 25 | | **Parameter** | **Locomotion** |
| Subsample $s$ | 16 | | $\lambda$ | 0.5 |

*Table 5.* Hyper-parameters used in the MetaWorld data scaling experiments.

## C. Data Scaling Results

Experiments for data scaling were conducted on the MetaWorld benchmark from Yu et al. (2020). Offline datasets for five different MetaWorld tasks were constructed as follows: Collect 100 trajectories of expert data on the target task using the built in ground truth policies with the addition of Gaussian noise of standard deviation 1.0. Collect 100 trajectories of sub-optimal data by running the ground-truth policy for a different randomization of the target task with Gaussian noise 1.0. Collect 100 trajectories of even more sub-optimal data by running the ground truth policy *of a different task* with Gaussian noise standard deviation 1.0 in the target domain. Finally, collect 100 trajectories with uniform random actions. As MetaWorld episodes are 500 steps long, this results in 200,000 time-steps of data. We then construct preference datasets by uniformly sampling segments from the offline dataset and assigning labels $y$ according to $\sum_t r(s_t^{(1)}, a_t^{(1)}) > \sum_t r(s_t^{(2)}, a_t^{(2)})$ where $r$ is the ground truth reward provided by metaworld. We then train using only the data from $\mathcal{D}$ General architecture hyper-parameters were taken from Lee et al. (2021a); Hejna & Sadigh (2022) which also use the MetaWorld benchmark, but for online preference-based RL. Full-hyper parameters are shown in Table 5. We run 20 evaluation episodes every 2500 steps. Full learning curves are shown in Figure 5. When reporting values in Table 2, we choose the maximum point on the learning curves which average across five seeds. This provides results as if early stopping was given by an oracle, which is less optimistic than averaging the maximum of each seed as done in Mandlekar et al. (2021).

## D. Ablations

In this section we provide additional ablations on both the benchmark datasets and MetaWorld datasets. We keep the hyperparameters the same, except for the parameter-efficient experiments. We run hyper-parameter sensitivty results for the human-preference benchmark datasets in Figure 6. The top row depicts the sensitivity for IPL to the value of $\lambda$. The bottom row depicts the sensitivity of MR to the number of timesteps the reward function is trained for.

For the parameter-efficient experiments *only* we use an efficient version of IPL based on AWAC (Nair et al., 2021) to additionally remove the value network. The outline of this variant is given below

---

**Algorithm 3** IPL Algorithm (AWAC Variant)

---

**Input:** $\mathcal{D}_p, \mathcal{D}_o, \lambda, \alpha$
**for** $i = 1, 2, 3, \dots$ **do**
    Sample batches $B_p \sim \mathcal{D}_p, B_o \sim \mathcal{D}_o$
    Estimate $V$ as $Q(s, \pi(s))$
    Update $Q$: $\min_Q \mathbb{E}_{B_p}[\mathcal{L}_p(Q)] + \lambda \mathbb{E}_{B_p \cup B_o}[\mathcal{L}_r(Q)]$
    Update $\pi$: $\max_\pi \mathbb{E}_{\mathcal{D}_p \cup \mathcal{D}_o}[e^{\beta(Q(s,a) - Q(s, \pi(s)))} \log \pi(a|s)]$

---

For this version of IPL, we use $\lambda = 0.5$. All other hyper-parameters remain the same as in Table 6 except the architectures. For the parameter-efficiency experiments only we use MLPs consisting of two dense layers with either dimension 64 or dimension 35. Running MR with a two-layer MLP of dimension 35 has almost exactly the same number of parameters as IPL-AWAC with two-layer MLPs of dimension 64. We include full results for the parameter-efficiency experiments in

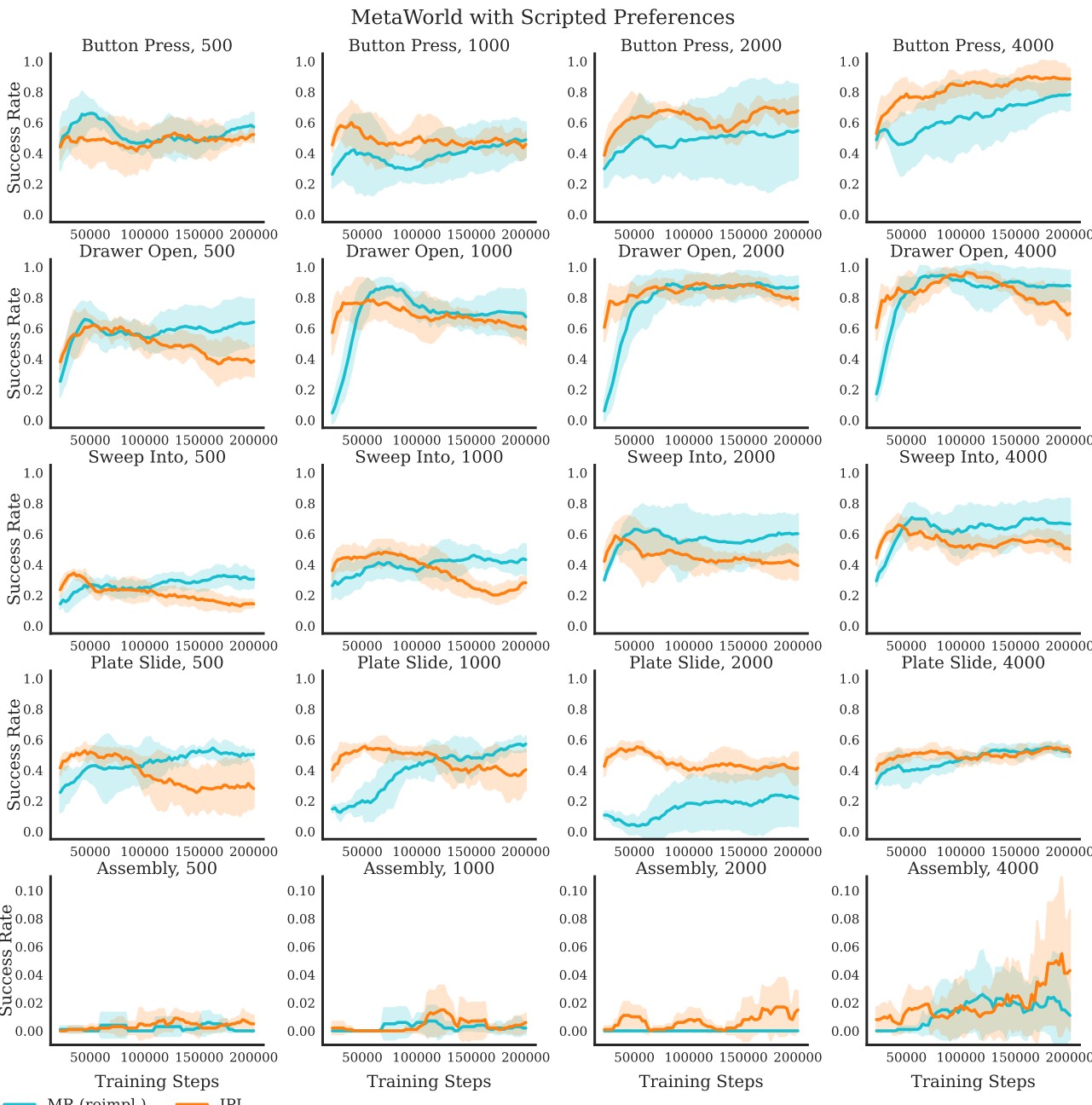

*Figure 5.* Full learning curves for the MetaWorld data scaling results with scripted preferences.

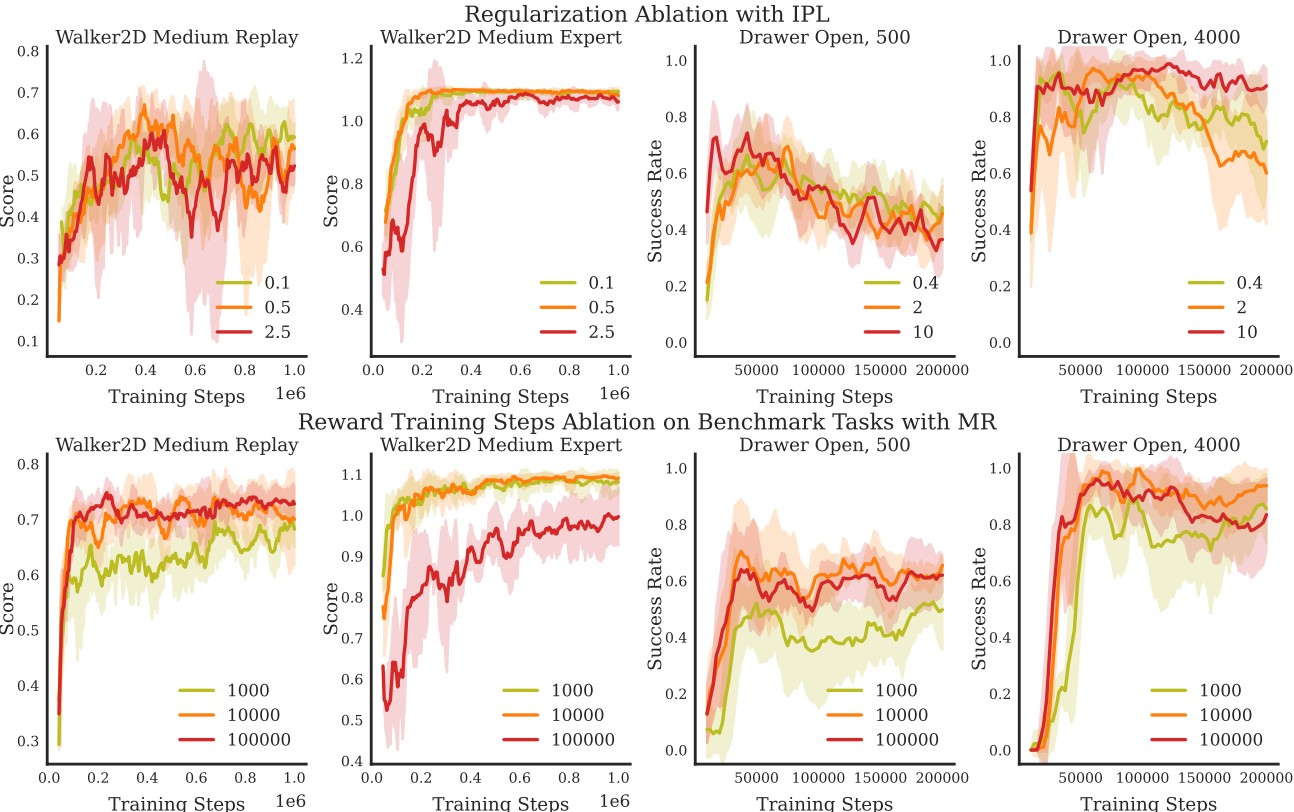

*Figure 6.* Ablations on regularization strength $\lambda$ for IPL (top row) and the number of reward steps for MR (bottom row). We see that IPL is relatively consistent across different values of $\lambda$. MR on the other hand, can vary greatly if the reward function under or over fits. In Walker2D Medium Replay and Drawer Open, 500, we see that it can easily under-fit. In Walker2D Medium Expert it easily over-fits.

| Preference Queries | | 500 | 1000 | 2000 | 4000 |
|---|---|---|---|---|---|
| | MR (35) | **73.9** ± 8.9 | **86.8** ± 8.2 | **89.9** ± 14.4 | **99.0** ± 1.0 |
| Button Press | MR (64) | 54.2 ± 16.1 | 42.6 ± 33.0 | 67.1 ± 14.9 | 43.4 ± 7.4 |
| | IPL (64) | 65.8 ± 13.3 | 79.8 ± 18.1 | 80.0 ± 17.3 | **95.8** ± 5.2 |
| | MR (35) | 13.4 ± 13.9 | 12.6 ± 21.9 | 15.5 ± 20.1 | 18.4 ± 25.6 |
| Drawer Open | MR (64) | 13.4 ± 19.0 | 57.1 ± 31.2 | 54.5 ± 31.7 | 78.8 ± 12.2 |
| | IPL (64) | **89.8** ± 11.3 | **93.2** ± 2.5 | **99.5** ± 0.9 | **95.5** ± 3.7 |
| | MR (35) | 35.1 ± 8.9 | 42.4 ± 9.9 | 45.9 ± 9.6 | 35.9 ± 4.1 |
| Sweep Into | MR (64) | 31.1 ± 6.4 | 55.8 ± 5.9 | 49.6 ± 10.3 | 56.4 ± 10.3 |
| | IPL (64) | **41.1** ± 14.2 | **63.9** ± 8.0 | **65.0** ± 12.0 | **63.9** ± 11.8 |
| | MR (35) | **55.2** ± 6.1 | **51.1** ± 4.4 | **53.0** ± 2.0 | 48.9 ± 3.3 |
| Plate Slide | MR (64) | 46.6 ± 21.9 | **50.8** ± 0.6 | 47.0 ± 2.5 | 48.5 ± 4.6 |
| | IPL (64) | **54.9** ± 3.2 | 49.4 ± 1.6 | 45.2 ± 9.0 | **48.8** ± 4.9 |

*Table 6.* Performance of different methods on the MetaWorld tasks under a limited parameter budget. MR (35) and IPL (64) have the same number of parameters. The Assembly task is ommited due to low success rate. On Button Press, fewer parameters appears to perform better as, due to the simplicity of the task, its easier for the bigger models to overfit. On Drawer Open and Sweep Into, we see consistent gains from increasing the number of parameters in the network, and IPL performs best overall. On the Plate Slide task, all methods at different parameter scales perform similarly.

Table 6. We find that on Drawer Open and Sweep Into, IPL outperforms both MR (64) and MR (35). In these environments, performance increases from MR (35) to MR (64) indicating that the expressiveness of the $Q$-function and policy are limiting performance. For the same budget, IPL is able to perform better. In Button Press, the simplest task, we find that MR (64) actually over-fits more than MR (35) and MR (64) ends up performing worse. In Plate Slide, all methods perform similarly independent of parameter count. We omit Assembly because of its low success rate at all data scales.

