# OpenReview forum: "Inverse Preference Learning: Preference-based RL without a Reward Function"
_ICML.cc/2023/Workshop/ILHF — ILHF Workshop ICML 2023_

### Official Review · Reviewer_WdnW · 2023-06-18
**Nice approach to go from preferences to policy without intermediate rewards, a more extensive empirical study would make the work stronger.**

**Rating:** 7
**Confidence:** 3

**Review:**

The paper presents a method for learning a policy from preferences in an offline scenario without the need to learn a reward model. The key idea is that the reward and Q functions can be used interchangeably, enabling policy learning through preferences to be expressed directly as a function of Q, which implicitly represents the reward function.

Pros:
A novel approach to policy learning from preference data that eliminates the need to learn an intermediary reward function. Through a number of experiments, the authors demonstrate that their method, IPL, surpasses existing baselines that learn from offline preferences across various tasks, such as locomotion and manipulation tasks in simulation.

Cons:
There seem to be missing gap between theory and practice where XQL is replaced by IQL in experiments. Is there a motivation for doing this? Also, it looks like there are limited experiments to convince the benefits of IPL over baselines. It would be great if a more extensive study can be done in light of limited theoretical understanding of the method. There are other relevant baselines that might be interesting to compare such as: SSSR (https://arxiv.org/abs/2010.11723), B-PREF(https://openreview.net/pdf?id=ps95-mkHF_), and ranking-game (https://arxiv.org/abs/2202.03481) and contrast against. Lastly, I think the rewriting reward implicit in Q has been studied in quite detail in a number of prior works before (ValueDICE and IQLearn) and might not be a contribution of this paper.

---

### Decision · Program_Chairs · 2023-06-20

Accept